# Oral Administration of TrkB Agonist, 7, 8–Dihydroxyflavone Regenerates Hair Cells and Restores Function after Gentamicin–Induced Vestibular Injury in Guinea Pig

**DOI:** 10.3390/pharmaceutics15020493

**Published:** 2023-02-02

**Authors:** Makoto Kinoshita, Chisato Fujimoto, Shinichi Iwasaki, Kenji Kondo, Tatsuya Yamasoba

**Affiliations:** 1Department of Otolaryngology and Head and Neck Surgery, University of Tokyo, Tokyo 113-8665, Japan; 2Department of Otolaryngology and Head and Neck Surgery, Nagoya City University, Nagoya 467-8601, Japan

**Keywords:** vestibule, semicircular canal, brain-derived neurotrophic factor, TrkB, 7, 8–Dihydroxyflavone, regeneration, caloric test, inner ear

## Abstract

The causes of vestibular dysfunction include the loss of hair cells (HCs), synapses beneath the HCs, and nerve fibers. 7, 8–dihydroxyflavone (DHF) mimics the physiological functions of brain-derived neurotrophic factor. We investigated the effects of the orally-administered DHF in the guinea pig crista ampullaris after gentamicin (GM)-induced injury. Twenty animals treated with GM received daily administration of DHF or saline for 14 or 28 days (DHF (+) or DHF (−) group; N = 5, each). At 14 days after GM treatment, almost all of the HCs had disappeared in both groups. At 28 days, the HCs number in DHF (+) and DHF (−) groups was 74% and 49%, respectively, compared to GM-untreated control. In the ampullary nerves, neurofilament 200 positive rate in the DHF (+) group was 91% at 28 days, which was significantly higher than 42% in DHF (−). On day 28, the synaptic connections observed between C–terminal–binding protein 2-positive and postsynaptic density protein-95-positive puncta were restored, and caloric response was significantly improved in DHF (+) group (canal paresis: 57.4% in DHF (+) and 100% in DHF (−)). Taken together, the oral administration of DHF may be a novel therapeutic approach for treating vestibular dysfunction in humans.

## 1. Introduction

Vestibular end organs in the inner ear, which detect linear and angular accelerations of the head, contribute to stabilizing gaze and posture. Various factors, such as aging, ototoxic drugs, or genetic disorders, can cause damage or loss of vestibular hair cells (HCs), thus resulting in vertigo, dizziness, oscillopsia, and postural instability [1,2,3,4,5,6,7]. While HCs can regenerate in both the auditory and vestibular systems in non–mammalian vertebrates [8,9,10,11], the level of spontaneous vestibular HC regeneration is very low in mammals [12,13,14,15]. Therefore, treatments for chronic vestibular dysfunction have been limited to date, especially in cases of bilateral involvement.

Neurotrophins are growth factors produced and secreted by neurons, glial cells, sensory cells, and muscle fibers. They play an essential role in the development and survival of neurons in the peripheral and central nervous systems [16,17]. For example, brain-derived neurotrophic factor (BDNF), a neurotrophin, exerts its physiological effects by activating the tropomyosin–receptor–kinase B (TrkB) receptors, thus regulating cell fate decisions, axon growth, dendrite growth, and the pruning of neurons during embryonic stages [18,19] and sustaining cell survival, morphology, and differentiation after birth [20].

Analyses of mice lacking the BDNF gene have shown that the survival of the primary vestibular neurons and their innervations in the vestibular system depend on BDNF [21,22]. Other studies have shown that mRNAs of BDNF and TrkB are expressed in the vestibular sensory epithelia and their afferent nerves in rats in both postnatal and adult stages [23] and that BDNF has promoting effects on neuronal survival and neurite outgrowth in the vestibular ganglion neurons of prenatal and newborn rats [24,25].

Because BDNF promotes neuronal survival and differentiation after birth, it has been used as a therapeutic agent for vestibular disorders in animal experiments. Previous studies have shown that BDNF effectively promotes neuronal survival and protects cells from ototoxic damage caused by cisplatin or gentamicin (GM) in the vestibular ganglion neurons of postnatal rats [26,27]. In addition, the administration of several growth factors, including BDNF, directly into the inner ear increased type I HCs in the vestibules of guinea pigs damaged by GM [28]. However, the direct administration of agents into the inner ear has a potential risk of causing inner ear damage, which greatly restricts its clinical applications.

7, 8–dihydroxyflavone (DHF) is a newly designed light molecule (254 Da) that can penetrate through the blood–brain barrier [29] and acts as a TrkB receptor agonist. DHF activates TrkB receptors and mimics the physiological functions of the cognate ligand BDNF [30], thus promoting adult neurogenesis. Intraperitoneal injection of DHF has been shown to treat cognitive deficits in mouse models of Alzheimer’s disease [31], prevent stress–induced memory deficits in rats [32], and be neuroprotective in a model of traumatic brain injury [33]. These findings suggest that DHF is a promising candidate for novel clinical therapeutic agent for neurodegenerative diseases. It is also important to note that DHF is orally bioavailable [34]; therefore, it can be dosed as an oral medication. 

Aminoglycoside antibiotics are the first ototoxic agents to highlight the problem of drug-induced hearing and vestibular loss [35]. Among them, it is well known that GM causes a greater degree of vestibulotoxicity than cochleotoxicity [36], and the HCs are the main targets of the damage [37]. GM has been used as an experimental model of peripheral vestibular dysfunction, and mild to moderate degeneration of the HCs has been shown in the cristae ampullaris (CAs) and the utricular macula in GM-treated guinea pigs [38,39].

In this study, we investigated the HC densities in the CAs, ampullary nerve densities, synaptic connection between the HCs and ampullary nerve endings, and vestibular function to assess the therapeutic potential of oral administration of DHF against GM-induced vestibular damage in the guinea pigs.

## 2. Materials and Methods

### 2.1. Animals

Twenty–five male Hartley guinea pigs, aged 4 weeks and weighing 250–300 g, were purchased from Tokyo Laboratory Animals Science Co., Ltd. (Tokyo, Japan). All animals were bred and housed in the standard animal facility under normal guinea pig rearing conditions, and experiments were performed according to guidelines of the University Committee for the Use and Care of Animals, University of Tokyo (approval number P18–002; Tokyo, Japan), as well as the National Institutes of Health Guide for the Care and Use of Laboratory Animals.

### 2.2. Experimental Protocol

On day 0, ototoxic GM (0.4 mg/mL, 0.2 mL) was injected into the left inner ear of 20 guinea pigs using cochleostomy under a surgical microscope (M320 F12, Leica, Germany) and general anesthesia with intramuscular ketamine 40 mg/kg and xylazine 10 mg/kg. The dose of GM in this experiment was determined according to our previous study in which we assessed the degree of HC damage in the CAs in guinea pigs [40]. This dose of GM damaged most HCs at 14 days after GM treatment and the extent of damage did not change at higher concentrations. The remaining five animals served as GM-untreated controls.

The GM-treated animals were divided into four groups. The DHF (+) group received daily oral administration of DHF (D1916, Tokyo Chemical Industry Co., Japan) at 19:00 for 2 weeks from day 1–14 or 4 weeks from day 1–28 after GM–induced injury (*n* = 5, each). The dose of DHF (5 mg/kg/day) in this experiment was determined according to a previous paper that showed this dosage was effective when used in vivo [34]. DHF (−) group received oral saline at 19:00 for 2 weeks from day 1–14 or 4 weeks from day 1–28 after GM–induced injury (*n* = 5, each).

Vestibular function was evaluated using caloric testing before GM administration and 14 days or 28 days after GM insult before the animals were euthanized for histological analysis (Figure 1).

### 2.3. Tissue Processing

The animals were euthanized with intramuscular ketamine 40 mg/kg and xylazine 10 mg/kg before and at 14 and 28 days after GM treatment. The collected left temporal bones were fixed in 4% paraformaldehyde/phosphate–buffered saline (PBS) (pH 7.4) for 2 h, and then the CAs were harvested under a microscope (SZX9, Olympus, Japan). The fixed specimens were immersed in PBS with 30% sucrose for 6 h, embedded in 5% agarose (type IX–A, Sigma–Aldrich, St. Louis, MO, USA) and 20% sucrose in PBS, then frozen in n–hexane (−60 °C). These specimens were cut vertically into 15 µm thick sections from the planum semilunatum to the center of the crista on a cryostat (Tissue–Tek Cryo3, Sakura Finetek, Tokyo, Japan) [41]. At intervals of 45 µm, five sections, including the center of CA, were immunostained.

### 2.4. Immunohistochemistry

Agarose-embedded cryosections of CAs were rinsed with 0.1 M PBS and then blocked with 5% goat serum in PBS with 0.1% Triton X–100 for 30 min at room temperature. For C-terminal-binding protein 2 (CTBP2) and postsynaptic density protein 95 (PSD–95) staining, sections were blocked and permeated with 5% goat serum in PBS with 4% Triton X-100 for 2 h. Then, the Image-iT FX signal enhancer (Thermo Fisher Scientific, Tokyo, Japan) was applied for 30 min at room temperature before the blocking process. The primary antibodies were monoclonal mouse anti-TrkB (1:100; R&D systems, Minneapolis, MN, USA), polyclonal rabbit anti–myosin 7a (MYO7A; 1:200; Proteus BioSciences, Ramona, CA, USA), monoclonal mouse anti–parvalbumin (PVALB; 1:100; Sigma–Aldrich, St. Louis, MO, USA), monoclonal mouse anti–neurofilament 200 (NF200; 1:100; Sigma Aldrich, St. Louis, MO, USA), rabbit anti–CTBP2 (1:100; Sigma Aldrich, St. Louis, MO, USA), and monoclonal mouse anti–PSD–95 (1:100; NeuroMab, Davis, CA, USA). Next, the antibodies were applied, and the sections were incubated for 24 h at 4 °C. After they were rinsed with PBS, the sections were incubated with Alexa Fluor fluorescent secondary antibodies (1:100; Molecular probes, Eugene, OR, USA) for 1 h at 37 °C. The secondary antibodies were goat anti–mouse Alexa488, goat anti-rabbit Alexa488, goat anti-rat Alexa488, goat anti-rabbit Alexa568, and goat anti-rat Alexa568. After rinsing with PBS, these sections were mounted in Vectashield Mounting Medium with 4′, 6-diamino-2-phenylindole (DAPI) (Vector Laboratories, Newark, CA, USA), and the slides were coverslipped. For negative controls, an absorption test was performed. These sections were observed under a confocal microscope (A1^+^, Nikon, Japan, ×40 or ×60 objective).

### 2.5. Confocal Microscopy

A series of images were obtained using a Nikon A1^+^ confocal imaging system. The laser power settings and pinhole were not varied for all the specimens. Amplifier offset and detector gains were optimized for each end organ to avoid fluorescence saturation of pixels, the state which an individual pixel is no longer able to store any more charge, in the images. Specimens were excited and recorded sequentially with the respective lasers to avoid cross-talk between different probes. Minimal adjustments to photodetector gain were necessary between different specimens to prevent saturation of pixels in any specimen; however, the gain was kept constant for all optimal image slices of a given tissue section.

### 2.6. HC Density

The HCs were identified by anti–MYO7A labeling of their cytoplasm. Type I HCs were identified by the basal location of their DAPI–labeled nuclei within the sensory cell layer and anti–PVALB labeling of their associated calyx afferent endings. Type II HCs were identified as the MYO7A–positive and PVALB–negative cells with DAPI–labeled nuclei at a more apical location within the sensory cell layer [42,43,44]. Unbiased investigators counted the number of HCs and supporting cells on the digital photomicrographs in five sections per animal. The basal lamina length was measured by tracing with a calibrated computer mouse (Photoshop CS5 software ver.12.0.4, Adobe Systems, San Jose, CA, USA). Each cell density was calculated by dividing the number of cells by the basal lamina length.

### 2.7. NF200 Positive Rate

For quantitative ampullar nerve analysis, immuno-stained images were obtained ×60 objective. The same image acquisition settings were used for samples from each of the different groups of animals analyzed. Z-stacks of optimal sections 0.5 µm thick were obtained, spanning 8 µm. After the banalization of images, the volume of the CA and NF200–labeled region was determined in three-dimensional (3D) reconstructions using the automatic 3D analysis module of NIS-Elements ver.4.3 (Nikon, Tokyo, Japan). Each NF200 positive rate was calculated by dividing the volume of the NF200–labeled region by the volume of the CA.

### 2.8. Synaptic Analysis

The number of Ctbp2 and PSD–95 puncta and the Ctbp2–PSD–95 colocalizations were determined in HCs to characterize synaptic elements. These quantitative data were obtained from all animals, from at least five cells per animal [45].

### 2.9. Caloric Test

The caloric test was used to evaluate the function of the lateral semicircular canal. After the normal condition of the tympanic membrane had been ascertained under the otoscope, both external meatuses were irrigated with 5 mL water at 0 °C for 20 s, with the animal lying prone with a maximal elevation of the head under dark conditions [46]. Then, the water was injected into the meatus using a 10 mL syringe and intravenous cannula to prevent any traumatic injury to the meatus or tympanic membrane. This procedure produces nystagmus directed away from the irrigated ear. A stopwatch and digital infrared charge-coupled device (CCD) video camera (DVSA10FHDIR, Kenko, Japan) were used to record the duration of the induced nystagmus. Quantitative data were derived using the standard Jonkee’s formula for canal paresis (CP) utilizing the duration of the induced nystagmus responses on the GM-treated and unaffected sides [47].

### 2.10. Statistical Analysis

SigmaPlot 11 statistical software (Systat Software Inc., San Jose, CA, USA) was used for statistical analysis, and all data were expressed as mean ± standard deviation. Mean HC densities, NF200 positive rates, the number of puncta per HC, and CP% were compared between groups. For these quantitative measures, multiple comparisons were made using a one-way analysis of variance (ANOVA) with Bonferroni post hoc test between control and DHF (+) or DHF (−) group at each time point. Statistical significance was set at *p* ≤ 0.05 after sequential Bonferroni adjustment for multiple tests. Mann-Whitney *U* test was used to determine the significance of differences between DHF (+) and DHF (−) group at each time point, and a *p* value of ≤0.05 was used.

## 3. Results

### 3.1. Increase in HC Densities by Oral Administration of DHF after GM Injury

Previously, we developed an animal model of acute vestibular injury in which guinea pigs received ototoxic GM administration using cochleostomy [40]. In this model, we confirmed that almost all HCs on the CA had disappeared 14 days after GM treatment. Furthermore, spontaneous HC regeneration of approximately half of the amount in GM-untreated controls was observed 28 days after GM treatment. Because of these findings, we adopted this animal model in this study.

To examine whether oral administration of DHF has effects on the recovery of the damaged vestibular HCs, GM-treated guinea pigs were divided into two groups. One group received oral administration of DHF continuously (DHF (+) group: *n* = 10), whereas the other group received oral saline (DHF (−) group: *n* = 10). GM-untreated controls (*n* = 5) possessed abundant HCs, divided into type I HCs labeled with PVALB and type II HCs, which are PVALB negative (Figure 2a). Almost all HCs disappeared in both DHF (−) and DHF (+) groups (*n* = 5 each, Figure 2b,d) 14 days after GM treatment; there was no difference in the number of the remaining HCs, type I HCs, and type II HCs between the groups. However, 28 days after GM treatment, the number of vestibular HCs in the DHF (+) group showed an increase of approximately 74% compared with the control levels. In contrast, the number of HCs in the DHF (−) group showed a recovery of approximately 49% compared with the control levels, with a significant difference between them (*p* < 0.01, Figure 2f). Furthermore, the type II HCs showed regeneration in both DHF (−) and DHF (+) groups, whereas the type I HCs regenerated only in DHF (+) group (Figure 2e); DHF (+) group showed a significantly greater increase in the number of both types I and II HCs than DHF (−) group (*n* = 5 each; Figure 2g,h; *p* < 0.01 and *p* < 0.05, respectively). These results demonstrated the high potency of DHF in promoting the regeneration of both types I and II vestibular HCs after GM–induced injury.

These results suggested that DHF promoted the HCs regeneration through TrkB in vestibular tissue based on previous reports that showed the high affinity of DHF for TrkB [29,30] and the effect of oral administration of DHF in animal models of central nervous system diseases [31,32,33]. However, the expression of most neurotrophins and their receptors is known to decrease to below detectable levels during adulthood in the inner ear [48]. Therefore, we examined the expression and the localization of the TrkB receptors in the vestibular endo-organs in adult guinea pig using immunohistochemistry. As a result, the expression of TrkB receptors was confirmed in the cytoplasm of HCs and the peripheral region of the ampullary nerves in guinea pigs (Figure 3).

### 3.2. Protection against Ampullary Nerve Damage by Oral Administration of DHF after GM Treatment

To examine whether the oral administration of DHF also protects the ampullary nerves and induces recovery, we stained the neurofilaments protein NF200 in peripheral ampullary nerve axons and compared ampullary nerve preservation between DHF (−) and DHF (+) groups. The ampullary nerves of the DHF (+) group were more well preserved than those of the DHF (−) group at both 14 and 28 days after GM–induced injury (Figure 4a–f).

Quantitatively, we compared the rate of the NF200–positive regions of each group with that of the GM–untreated controls using three-dimensional images provided by Z–stack photography. NF200–positive rates in the DHF (+) group were significantly higher than those in the DHF (−) group at both 14 and 28 days after GM treatment (Figure 4f; *p* < 0.01 and *p* < 0.05, respectively). However, NF200–positive rates did not change from 14–days after GM treatment in either group. These results suggest that DHF effectively protects the ampullary nerves from GM-induced damage but does not enhance their recovery.

### 3.3. Synaptic Change in Type I HCs by Oral Administration of DHF after GM–Induced Injury

To examine whether DHF administration also influences synaptic remodeling after GM-induced injury, we labeled vestibular cristae with antibodies against CTBP2, a marker of presynaptic ribbons in HCs, and PSD–95, a marker of postsynaptic densities, in peripheral ampullary nerve endings. In GM-untreated controls, many PSD-95-positive and CTBP2–positive puncta were observed in HCs (Figure 5a). PSD-95 positive puncta were rarely observed in the HCs 28 days after GM-induced injury in the DHF (−) group; however, some CTBP2 positive puncta were observed (Figure 5b). Furthermore, many PSD-95-positive puncta and CTBP2-positive puncta were observed in HCs 28 days after GM-induced injury in the DHF (+) group (Figure 5c); colocalization of CTBP2 and PSD–95 was also observed.

The number of PSD-95 puncta per HC was significantly decreased (*p* < 0.01) in DHF (+) and DHF (−) groups than in controls; however, it was significantly greater in the DHF (+) group than in the DHF (−) group (*p* < 0.01, Figure 5d). While the number of CTBP2–positive puncta per HC was reduced in DHF (−) and DHF (+) groups, the difference was insignificant among the controls, DHF (−), and DHF (+) groups (*p* > 0.05). In addition, the number of puncta showing the colocalization of CTBP2 and PSD–95 was significantly greater in the DHF (+) group than in the DHF (−) group (*p* < 0.05, Figure 5e). These results suggest that DHF promotes synaptic remodeling between the HCs and ampullary nerve endings.

### 3.4. Recovery of Vestibular Function by Oral Administration of DHF after GM–Induced Injury

Finally, to examine whether DHF can promote the recovery of vestibular function, we performed caloric testing in both ears 14 and 28 days after GM–induced injury. Before GM injection, there were no significant differences in the duration of caloric nystagmus between the DHF (+) and DHF (−) groups. No nystagmus was observed in the DHF (−) group 14 days after GM–induced injury; however, weak nystagmus was elicited in the DHF (+) group. While the extent of canal paresis was not significantly different between groups, the DHF (−) group still showed no nystagmus 28 days after GM injury; however, stronger nystagmus was elicited in the DHF (+) group. Furthermore, the canal paresis was significantly reduced (*p* < 0.01) from 14 to 28 days after GM treatment in the DHF (+) group and was significantly smaller in the DHF (+) group than in the DHF (−) group 28 days after GM treatment (*p* < 0.01, Figure 6). These results suggest that oral administration of DHF induces a functional recovery of the peripheral vestibular system, which corresponds well with the histological findings, including the enhanced regeneration of the HCs, protection of the ampullary nerves, and promotion of their synapses.

## 4. Discussion

In this study, we examined the effects of the orally administered TrkB agonist, DHF, on GM-induced vestibular damage in guinea pigs and showed that DHF could promote HC regeneration, vestibular nerve preservation, and synaptic remodeling, thus resulting in partial recovery of vestibular function.

While non-mammalian vertebrates can spontaneously regenerate HCs in the inner ear after HC damage by ototoxic drugs or acoustic trauma [8,9,10,11], HCs in the mammalian inner ear have a very limited ability to regenerate after damage, leading to permanent hearing and balance impairments [13,14,15,49,50,51]. Furthermore, while the vestibular end organs in mammals have some ability to regenerate spontaneously after ototoxic damage [13,15,40,52], the quantity and quality of the spontaneously regenerated HCs are not sufficient to restore balance after damage to the inner ear [53].

Various growth factors have been studied to determine if they can promote the spontaneous regeneration of HCs in mammalian vestibular end organs [27,28,44,54,55,56,57,58,59]. Studies that used utricular cultures obtained from rats and mice have shown that transforming growth factor α (TGFα), epidermal growth factor (EGF), fibroblast growth factor (FGF), and insulin–like growth factor–1 (IGF–1) can all promote the proliferation of vestibular HCs [44,58]. In addition, the administration of TGFα, IGF–1, or BDNF directly to the inner ear has also been shown to enhance vestibular HC regeneration after ototoxic damage in rats and guinea pigs [28,54]. However, efficient delivery of these agents to the inner ear is problematic because the direct administration of these agents with large molecular weights to the inner ear has the potential risk of causing permanent hearing loss.

DHF is a member of the flavonoid family of compounds in fruits and vegetables [30]. The binding of DHF to the cysteine cluster 2 and leucine-rich region in the extracellular domain of the TrkB receptor provokes the receptor’s dimerization and autophosphorylation, leading to the activation of downstream signaling cascades similar to BDNF. In addition to its potential as a useful clinical treatment, DHF is orally bioactive and can penetrate the blood–brain barrier [34]. Furthermore, DHF has been shown to have neuroprotective effects against oxidative stress incurred from glutamate-induced neurotoxicity [60], decreases infarct volume in stroke [61], protects against traumatic brain injury [33], treats cognitive deficits in mouse models of Alzheimer’s disease [62], and protects against degeneration in spiral ganglion neurons in congenitally deaf mice [63].

Regarding the effect of BDNF on vestibular HCs, previous studies have shown that BDNF is not critical for their survival [18,22] and does not protect them from ototoxic drugs [44]. In the present study, almost all of the vestibular HCs at the CA had disappeared 14 days after GM treatment in both DHF (−) and DHF (+) groups, which is consistent with the results of previous studies. However, vestibular HC regeneration was significantly increased in the DHF (+) group than in the DHF (−) group 28 days after GM treatment, suggesting that orally administered DHF has a promoting effect on HC regeneration. Among previous studies of ototoxin–induced vestibular injury, some without trophic factor treatment showed spontaneous regeneration of type II vestibular HCs, but virtually no recovery of type I HCs in guinea pigs and chinchillas [64,65,66]. In contrast, other studies with transotic administration of BDNF showed an increase in type I HCs in the GM-damaged vestibules of chinchillas [28,56]. The present study has also shown that oral administration of DHF led to massive regeneration of the type I HCs in CA after GM–induced damage.

Regarding the effect of BDNF on vestibular ganglion neurons, previous studies have reported that BDNF is an essential survival factor in vivo [21,22] and in vitro [27] and can protect vestibular ganglion neurons from ototoxic drugs in vitro [26,27]. In the present study, the ampullary nerves were well preserved in the DHF (+) group than in the DHF (−) group at both 14 and 28 days after GM treatment, suggesting that orally-administered DHF promoted the preservation of the ampullary nerves. The HCs in a healthy inner ear provide neurotrophins, including BDNF, to the afferent neurons [67,68]. Therefore, a secondary problem that emerges from the loss of HCs is that the afferent neurons lose their neurotrophic support, thus resulting in their gradual degeneration [48]. Recently, it has been reported that BDNF promotes the elongation of regenerating axons [69,70] and that DHF also promotes axonal regeneration in truncated peripheral nerves [71]. Therefore, ampullary nerve preservation observed in the present study may result from the compensatory BDNF-like effect of DHF after the loss of the HCs.

Because expression of the TrkB receptor was abundant in HCs and ampullary nerves, DHF was expected to protect HCs against GM-induced HC loss. However, HCs remarkably decreased in DHF (−) and DHF (+) groups 14 days after GM–induced injury. It is presumed that because the first administration of DHF was one day after GM injection, many HCs suffered fatal damage during that time.

While the regeneration capacity of the vestibular HC persists to a limited extent in some mammalian species [13,15], sufficient functional recovery is not achieved [1,72,73]. Regeneration of synaptic connections between the HCs and vestibular ganglion neurons is required to restore vestibular function after spontaneous HC regeneration [45]. In the vestibular sensory epithelia, the presynaptic active zones contain ribbons characterized by the ribeye/Ctbp2 core protein. In contrast, the calyces contain postsynaptic densities rich in PSD–95, where the glutamate alpha-amino-3-hydroxy-5-methyl-4-isoxazolepropionic acid(AMPA) receptors are clustered [74]. BDNF is an essential trophic molecule for vestibular ganglion neurons [22] and has been suggested to play a role in differentiating calyceal afferents on type I vestibular HCs [75]. DHF mimics the physiological functions of BDNF, including promoting neuronal survival and elevating synaptogenesis [30,76,77,78,79]. In the present study, the synaptic connections between presynaptic ribbons and postsynaptic densities were restored in the DHF (+) group, and the caloric response was significantly improved in the DHF (+) group than in the DHF (−) group. These results suggest that DHF restored the synaptic connections, thus contributing to the recovery of vestibular function after vestibular HC regeneration.

In conclusion, the promoted HC regeneration, vestibular nerve preservation, and synaptic remodeling induced by oral administration of DHF are considered to result in the recovery of vestibular function after GM treatment (Figure 7). While further studies are required to investigate the activity of the BDNF–TrkB pathway with/without DHF after GM-induced vestibular injury, this study’s results are an important and logical starting point for the effective treatment of vestibular diseases.

## Figures and Tables

**Figure 1 pharmaceutics-15-00493-f001:**
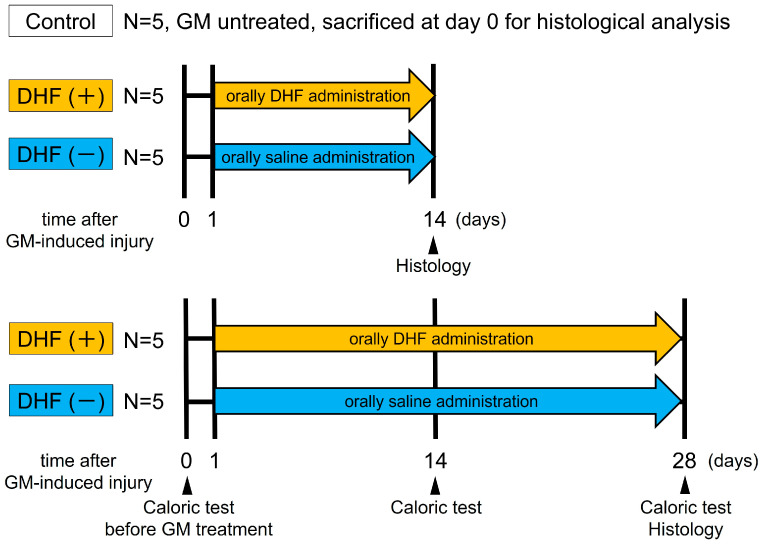
A schematic representation of the experimental procedure. Five animals served as GM–untreated histological controls. The GM–treated animals were divided into four groups. DHF (+) group received daily oral administration of DHF from day 1–14 or day 1–28 (*n* = 5, each) after GM–induced injury. DHF (−) group received oral saline after GM-induced injury (*n* = 5, each). DHF, 7, 8–Dihydroxyflavone; GM, gentamicin.

**Figure 2 pharmaceutics-15-00493-f002:**
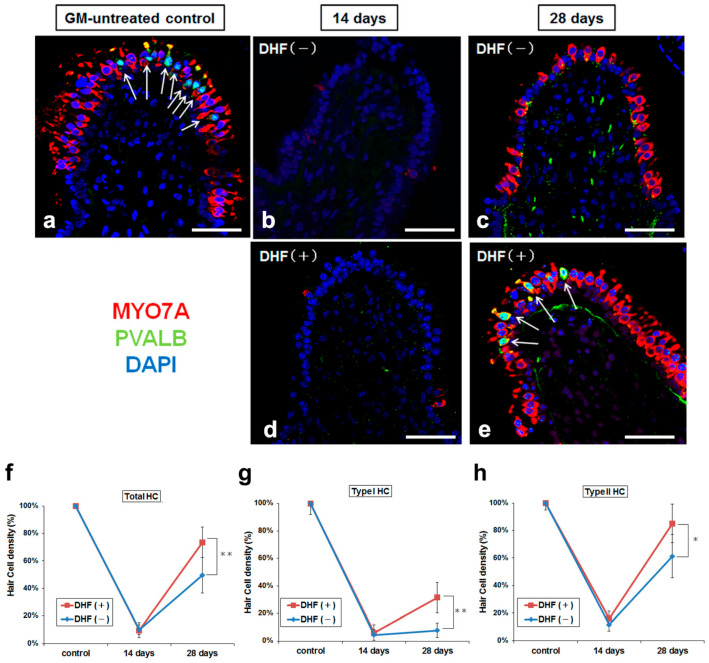
Changes in HC densities. (**a**–**e**) Confocal images of crista ampullaris stained with MYO7A antibody (red), PVALB antibody (green), and DAPI (blue). (**a**) GM-untreated control shows abundant HCs in the sensory epithelium and many type I HCs, double-stained with MYO7A and PVALB (arrow). (**b**,**d**) The HCs are remarkably decreased in both DHF (−) and DHF (+) groups 14 days after GM treatment. (**c**,**e**) The HCs are recovered in both groups 28 days after GM treatment. There are some type I HCs in DHF (+) group. (**f**–**h**) Changes in the densities of total HCs, type I HCs, and type II HCs after GM treatment with/without orally administered DHF (N = 5 for control, DHF (+) and DHF (−) at 14 and 28 days). Type I HCs almost disappeared in both groups 14 days after GM treatment, whereas type Ⅱ HCs decreased in both groups. Both types I and II HCs are significantly increased in the DHF (+) group than in the DHF (−) group. Scale bar (**a**–**e**), 50 µm; DAPI, 4′, 6-diamino-2-phenylindole; DHF, 7, 8–Dihydroxyflavone; GM, gentamicin; HC, hair cell; MYO7A, myosin 7a; PVALB, parvalbumin; ** *p* < 0.01; * *p* < 0.05 (Mann-Whitney *U* test).

**Figure 3 pharmaceutics-15-00493-f003:**
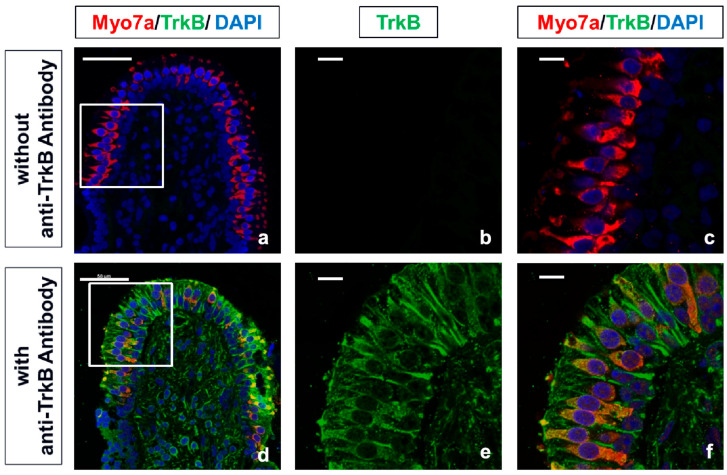
Confirmation of the TrkB receptor expression in crista ampullaris. (**a**–**c**) Upper panels are negative controls immunostained without the anti–TrkB antibody. (**d**–**f**) Lower panels show immunostaining with anti-TrkB antibody. Each middle and right panel (**b**,**c**,**e**,**f**) magnifies the boxed area in the respective left panel (**a**,**d**). Red represents MYO7A–labeled HCs, green represents TrkB receptors, and blue represents DAPI-labeled nuclei of cells. Expression of the TrkB receptor is confirmed in the cytoplasm of HCs and the peripheral region of the ampullary nerves. Scale bars (**a**,**d**), 50 µm; (**b**,**c**,**e**,**f**), 10 µm; DAPI, 4′, 6-diamino-2-phenylindole; HC, hair cell; MYO7A, myosin 7a; TrkB, tropomyosin–receptor–kinase B.

**Figure 4 pharmaceutics-15-00493-f004:**
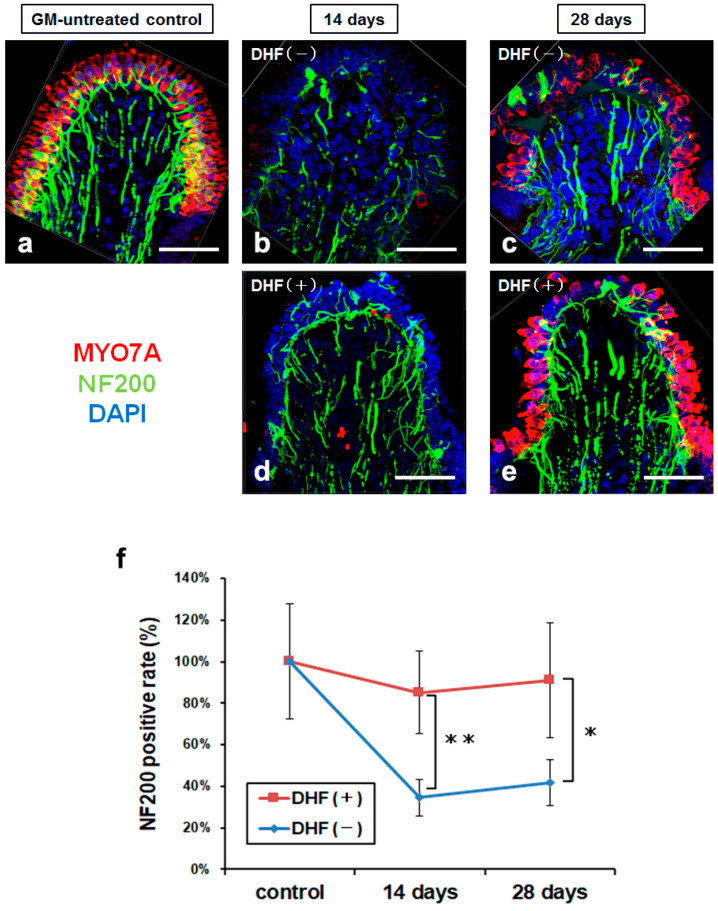
Protection of nerve fiber. (**a**–**e**) Three-dimensional images of the crista ampullaris stained with MYO7A antibody (red), NF200 antibody (green), and DAPI (blue). (**a**) GM-untreated control. (**b**,**c**) The cristae without DHF at 14 and 28 days after GM injection. (**d**,**e**) The cristae with DHF at 14 and 28 days after GM injection. The ampullary nerves of the DHF (+) group are better preserved than those of the DHF (−) group at 14 days and 28 days after GM treatment. (**f**) NF200 positive rates are significantly higher in the DHF (+) group than in DHF (−) group at both 14 and 28 days after GM treatment. N = 5 each for control, DHF (+) and DHF (−) groups at 14 and 28 days. Scale bar (**a**–**e**), 50 µm; DAPI, 4′, 6-diamino-2-phenylindole; DHF, 7, 8–Dihydroxyflavone; GM, gentamicin; MYO7A, myosin 7a; NF200, neurofilament 200; ** *p* < 0.01; * *p* < 0.05 (Mann-Whitney *U* test).

**Figure 5 pharmaceutics-15-00493-f005:**
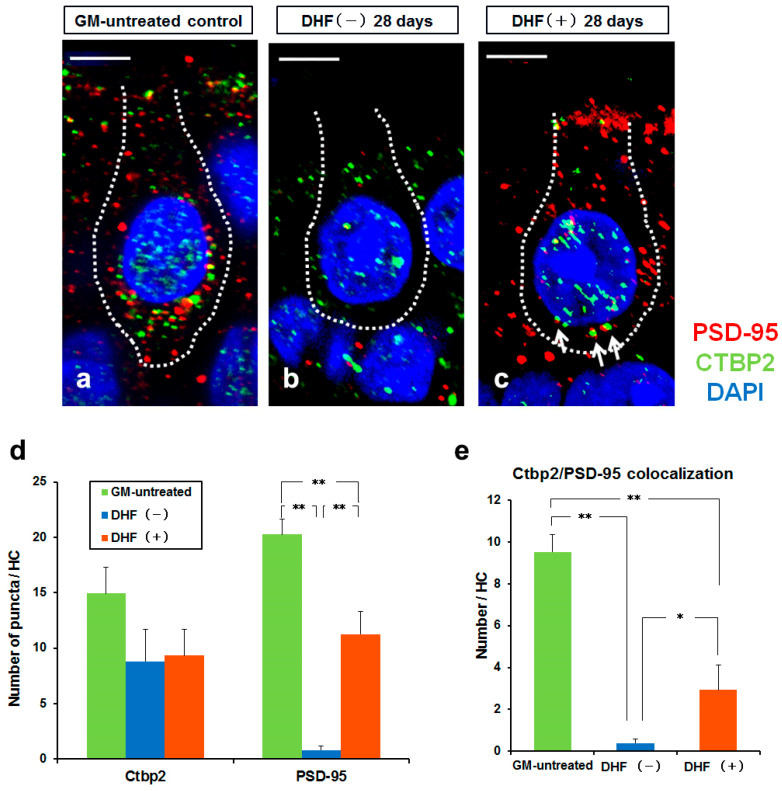
Synaptic analysis. (**a**–**c**) Confocal images of the crista ampullaris stained with PSD–95 antibody (red), CTBP2 antibody (green), and DAPI (blue). The dotted lines show the outline of HCs. (**a**) The GM–untreated controls have many PSD–95 positive and CTBP2 positive puncta in the HC. (**b**) In the DHF (−) group, PSD-95 positive puncta were not observed 28 days after GM injection. (**c**) In the DHF (+) group, many PSD-95 positive puncta, some CTBP2 positive puncta, and some CTBP2/PSD–95 colocalizations (arrow) are observed in the HC 28 days after GM injection. (**d**,**e**) N = 5 each for control, DHF (+) and DHF (−) groups at 14 and 28 days. (**d**) The number of CTBP2 positive puncta per HC is not significantly different among groups. The number of PSD-95 puncta per HC is significantly decreased in the DHF (−) group. (**e**) CTBP2/PSD–95 colocalization is significantly greater in the DHF (+) group than in the DHF (−) group. Scale bar (**a**–**c**), 5 µm; CTBP2, C–terminal–binding protein 2; DAPI, 4′, 6-diamino-2-phenylindole; DHF, 7, 8–Dihydroxyflavone; GM, gentamicin; HC, hair cell; PSD–95, postsynaptic density protein 95; ** *p* < 0.01; * *p* < 0.05, tested by one-way ANOVA and Bonferroni post hoc test.

**Figure 6 pharmaceutics-15-00493-f006:**
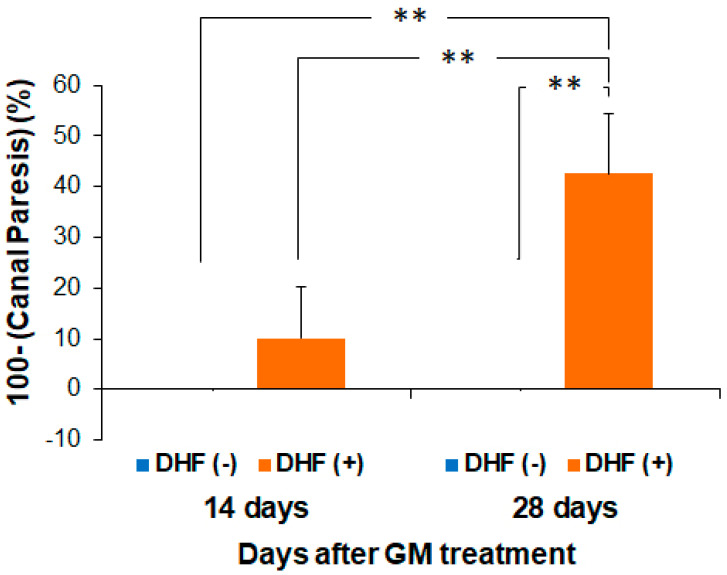
Evaluation of vestibular function. The DHF (−) group showed no nystagmus 14 or 28 days after GM–induced injury. In addition, the canal paresis was significantly improved in the DHF (+) group than in the DHF (−) group. N = 5 each for control, DHF (+) and DHF (−) at 14 and 28 days. DHF, 7, 8–Dihydroxyflavone; GM, gentamicin; ** *p* < 0.01, tested by the Mann-Whitney *U* test by comparing with DHF (+) at 28 days.

**Figure 7 pharmaceutics-15-00493-f007:**
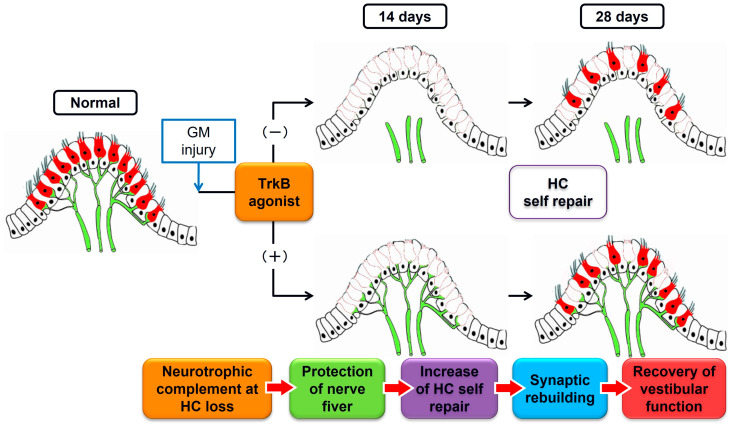
Schematic representations of the effect of oral TrkB agonist administration.

## Data Availability

The data presented in this study are available in the article text.

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
