# Peer review of "Oral Administration of TrkB Agonist, 7, 8–Dihydroxyflavone Regenerates Hair Cells and Restores Function after Gentamicin–Induced Vestibular Injury in Guinea Pig"

_pharmaceutics, 2023, doi:10.3390/pharmaceutics15020493_

Round 1

Reviewer 1 Report

Lack of robust regeneration in mammalian inner ear leads to permanent dysfunction of auditory and vestibular organs, and new drugs and treatments are needed to promote hair cell regeneration and synapsis re-establishment for function restoration. This manuscript “Oral administration of TrkB agonist, 7, 8–dihydroxyflavone, re-2 stores vestibular function after gentamicin–induced hair cell 3 loss in guinea pig” describes the regenerative and protective effect of DHF after gentamicin-induced hair cell damage in the guinea pig crista ampullaris. The authors compared the organs from DHF treated animals to that from control groups by immunostaining of hair cells, ampullary nerves, pre- and post-synaptic complexes, as well as functional recoveries at different timepoints. The authors found that DHF treatment not only promotes hair cell regeneration in crista ampullaries, but also protects ampullary nerves and facilitates the synaptic connections. Most importantly, they confirmed that DHF treatment helps the functional recovery of the vestibular system, providing a potential therapeutical drug. This study was well designed, and comparisons were carefully performed; the manuscript was well written and easy to understand. There are a few minor issues in the manuscript.

11)    Should clearly state in the abstract that this study was focused on crista ampullaries.

22)      Section 2.2, line 89 and line 93, “oral administration of DHF at 19.00”, does “19.00”mean 19:00 or 7:00pm? Please clarify.

33)      The n for each comparison was not clearly stated. For example, how many organs were used for hair cell staining to compare DHF+ vs DHF-? Please include n for figure 2~6 in the figure legends.

44)      Figure 2b and 2d, are these figures representative? There are more type II hair cells in DHF+ organs in 2d than 2b, but the quantification in 2f and 2h show no difference between DHF+ and DHF-.

5)5      Last paragraph of section 3.1, positive immunostaining signals for TrKB receptors in hair cells and ampullary nerves is not sufficient for the confirmation of TrKB receptor activation. Either provide immunostaining of phosphorylated-TrkB receiptor to show TrkB activation, or rephrase these sentences.

In addition to above issues, I am curious whether any comparison was done for utricles or cochleae. If yes, it might be better to include a paragraph to discuss the results for the interest of general audience.

Reviewer 2 Report

Dear Ladies and Gentlemen, Dear Journal-Team,

the manuscript 'Oral administration of TrkB agonist-, 7,8-dihydroxyflavone, restores vestibular function after gentamicin-induced hair cell loss in the guinea pig' is well written. The figures are sufficient and illustrative.

1. For the calculation of the needed animals the control group can be smaller or only few animals.

2. What do you mean by saturation of pixels (section 2.5 Confocal microscopy): colour units per area or loaded particles per area in eV or difference of loaded particles in adjacent areas?

3. Please explain the abbreviations Ctbp2 (C-terminal binding protein 2) and PSD (postsynaptic density) should be already in the abstract. Explain the fluorescent stain DAPI (4,6-diaminodino-2-phenylindole), when first used in the section Immunohistochemsitry, and in Figure 2 as well.

4. Minor points: Please check for spacing in the second line from below of the abstract. Change in the legend of Figure 1 to 'day 1-28'. Change in Figure 3 to 'Protection of nerve fibre'.

5. Please check the references for continuous numbering and check Reference 72 by Tian et al. for spacing.

Sincerely,

Reviewer 3 Report

Paper titled (Oral administration of TrkB agonist, 7, 8–dihydroxyflavone, restores vestibular function after gentamicin–induced hair cell loss in guinea pig) by Kinoshita et al. tested the regenerating role of TrkB agonist on vestibular function in guinea pigs received gentamycin. 

1- Title: better to be modified to be more informative. What was this impact of oral administration? whether increase or decrease in function or structure?

2- Abstract should be amended by some numerical values.

3-Introduction: give a brief account on gentamycin & its uses as this is the main issue in your research

4-Define each abbreivation at the first appearance

5- The aim at the end of introduction looks like a conclusion. Here authors should summarize their aim and how they achieved it. So this part should be rewritten.

6- Write the source from which you purchased the animals

7- Add refernces for the doses of GM & Trk agonist

8- Figure 4: the SD value is cut at the top of the curve

9- Figure 6: are there zero values? if so, start the x axis below zero to show them

10- For statistical analysis, authors should mention that they compared each set separately by ANOVA.
Really it is NOT clear how authors applied one-way ANOVA on these different groups at multiple time points. 
Authors should consult a statistician to advice on statistical analysis and confirm for this in their reply & show the method of stat analysis clearly in METHODS & figure legends

11- mention "n" in each figure legend

Round 2

Reviewer 3 Report

thanks